# Social Acceptability of More Sustainable Alternatives in Clothing Consumption

**Silke Kleinhückelkotten * and H.-Peter Neitzke**

ECOLOG-Institut für Social-Ökologische Forschung und Bildung, 30449 Hannover, Germany;
peter.neitzke@ecolog-institut.de
* Correspondence: silke.kleinhueckelkotten@ecolog-institut.de

 

**Abstract:** The rapidly growing demand for clothing in connection with the resource requirements and the emissions along the textile chain as well as the prevailing working conditions in the textile industry cause serious environmental and social problems. The question is asked, whether changes in consumption towards more sustainably produced clothing and, finally, a reduction of clothing consumption are achievable against the background of the existing consumption-related patterns of attitudes and behaviors. A representative survey was conducted in Germany ($N$ = 2000) to tackle the consumer-related aspects of this question. The characteristics of consumption-related attitudes in the different population segments were determined. Factors were identified that affect the buying and use of clothes as well as the efficiency, consistency, and sufficiency supporting consumption alternatives. The results show that some preconditions for a broader diffusion of more sustainable alternatives in clothing consumption are given in Germany, such as a widespread general sustainability and problem awareness. In some population segments, social norms supporting more efficiency and consistency in the clothing sector are effective, and social and ecological buying criteria have a relatively high importance. However, there are also strong attitudinal obstacles, particularly regarding the restriction of clothing consumption.

**Keywords:** clothing consumption; efficiency; consistency; sufficiency; attitudes; buying behavior; representative survey

## 1. Introduction

Worldwide, the demand for clothing increases much faster than the population [1]. Reasons for this development are high levels of consumption in the economically developed countries, enlivened by frequent changes of fashion, and also rising incomes in the threshold economies. This is problematic because the conventional textile chain is highly unsustainable from the provision of the raw materials and the manufacture of the garment through the use phase to the disposal of clothing waste. The dominating production processes for and the use of clothing, as well as recycling and disposal, not only cause serious environmental problems but are also connected with health risks, poor working conditions, and other adverse social effects [2–4].

In the context of clothing, sustainability means that negative ecological impacts as well as impairments of the living conditions of workers, users, and all parties affected in any manner are precluded, both now and in the future and for the entire textile chain, that is, in all phases of the production, use, reuse, and recycling of clothes as well as in the treatment and handling of clothing waste.

Generally, three strategies can contribute to improve sustainability in the production and consumption of any product [5–7]:

- Efficiency: Minimize the input of resources (land, energy, raw materials) to produce the consumer good and use it intensively.
- Consistency: Substitute resources and adapt production processes to natural resource flows to produce products in a more environmentally friendly manner.
- Sufficiency: Restrict consumption to a level that is enough for a healthy and satisfactory life but avoids excess.

A good deal of the responsibility to improve sustainability in the clothing sector lies with the textile industry. There are many options to enhance efficiency and consistency along the textile chain [8–11]. However, changes in consumer behavior and more sustainable consumption patterns are also essential. Sustainable consumption denotes the use of goods and services that satisfy basic needs and facilitate a certain quality of life, while minimizing environmental and social harm to present and future generations. On the part of the consumers, efficiency and consistency in the textile sector could be supported and pushed by:

- prolonging of the useful lifetime of clothing, e.g., by choosing high-quality garments, by repairing shopworn clothes, by upcycling of used clothes or by passing them to others;
- buying of timeless instead of voguish clothing, more sustainably produced clothing, second-hand instead of new clothes, or clothing made from recycled parts or fibers;
- sharing, swapping, lending, renting, or leasing of clothing instead of buying it;
- laundry with reduced temperatures and environmentally compatible detergents;
- consigning sorted out clothes to recovery.

Sufficiency, in the end, comes down to a willful buying resistance aiming at the restriction of the number of clothing items to a minimum or at least to clothes with a positive environmental and social balance. The longer use of clothing could contribute not only to efficiency but also to sufficiency if it is backed by consumer reticence. Such ultimately non-consumption practices are the core element of what is called 'strong sustainable consumption', as compared against 'weak sustainable consumption' which solely is directed to improvements of material, social, and institutional efficiency [12–14].

In several studies for different countries the importance attached to clothing, the volume of clothing purchases, the spending for clothing, the preferred sources of supply, the useful life of clothing, the reasons for the sorting out of clothes, and the handling of sorted out clothes were investigated. The results of the representative surveys showed:

- The spending for clothing, the purchase volume, and the clothing stock of people with high incomes and high educational attainment are significantly above average (Germany, Poland, Sweden, U.S.A. [15]; Germany [16,17]).
- The main motives for the purchase of clothes are replacement needs (28.1%), desire for something new (17.5%), special offers (16.0%), and impulse buying (12.5%) (Germany [18]).
- There are practically no differences between different consumer segments with respect to the time clothes are kept before being discarded and the frequency of wearing each clothing item (Germany, Poland, Sweden, U.S.A. [15]).
- Only 21% of the consumers sort out clothing solely because it is worn out or does not fit anymore (Germany [17]).

The results of two of the already mentioned and some other representative studies allow conclusions with respect to the prevalence of and the diffusion potential for more sustainable clothing consumption behaviors. The following findings are noteworthy:

- For about 40% of German consumers, social and environmental compatibility of the production of clothing are relevant buying criteria (Germany, United Kingdom, France, Italy, Spain, [19]).

- The percentage of consumers buying clothes made from organic materials increases with income, i.e., it increases from the low-budget to the high-premium consumer segment (Germany, Poland, Sweden, U.S.A. [15]).
- The willingness to buy clothes produced under environmentally and socially acceptable production conditions is higher among women compared to men and increases with age (Germany [17]).
- The value consumers attribute to second hand-clothing and clothing made of recycled material is lower than for conventional clothing. Compared to clothing made of conventional materials, that made of organic materials is rated higher in the high casual and premium consumer segments but lower in the low-budget and low-casual segments (Germany, Poland, Sweden, U.S.A. [15]).
- For less than 10% of the consumers, the buying of second-hand clothing is a real option (European Union [20]).
- The buying of second hand-clothing is more widespread among women compared to men and can be found more frequently in younger than in older population segments (Germany [17]).

For many products the decision in favor or against more or less sustainable consumption options is governed by a broad spectrum of cultural, social, personal, situational, as well as product-related factors [21–23]. The importance of the purchase situation and the product characteristics have been objects of many, mainly marketing-oriented, studies (e.g., [24–27]). In studies with a psychological focus the influence of, e.g., values, norms, needs, and emotions, on intentions and behaviors supporting a more sustainable clothing consumption were analyzed. The works are often based on the theories of planned behavior [28] and/or norm activation [29] and extensions of these theories. In the context of the present study the following findings from recent studies are interesting:

- Biospheric and altruistic values have a positive effect, while egoistic and hedonic values have a negative effect on attitudes towards more sustainably produced clothing (Germany; 1085 female consumers with a certain openness to the purchase of sustainable clothing in the middle- to high-prize segment [30]).
- The consideration of sustainability criteria in buying decisions for clothes is positively influenced by biospheric values and the feeling of compassion for vulnerable others, while there is a negative relation to hedonic values (Germany; 981 consumers all from the same small town [31]).
- Altruistic values have a positive effect on attitudes towards collaborative fashion consumption and the corresponding behavior. The association with egoistic values is slightly negative. There is also a negative effect of age on the behavior. All effects are quite small (1014 consumers all from the same small town [32]).
- There is a positive relation between personal norms and the intention to not consume clothing deemed problematic and to consume less clothing. The personal norms are associated with problem awareness, ascription of responsibility, and outcome efficacy (USA, Germany, Sweden, Poland; 4591 consumers from the general population [33]).
- Anticipated guilt is a major driver for fair-trade buying behavior (USA; 430 consumers [34]).
- Fashion leadership is positively associated with the intention to participate in clothing renting and swapping. The same relation can also be found for the need for uniqueness and the swapping intention. Materialism is negatively related to both renting and swapping (35: USA; 431 females [35]).

There are some more studies but most of them exhibit methodic weaknesses, as also can be found in some of the cited studies, like the selection of very special test groups, e.g., students or users of certain shopping offers, often combined with low numbers of cases. In the above literature survey studies made against a completely different cultural background than that of the area under investigation of this study have been excluded.

This paper deals with the question to what extent the existing consumption-related patterns of attitudes and behaviors support or impede the diffusion of more sustainable consumption alternatives

and thus the implementation of the three sustainability strategies in the field of clothing. In detail the research questions are:

(1)　Are there social differences with respect

　　　(a)　to the expression of attitudes and the relevance of social norms related to clothing consumption and
　　　(b)　the consumption behavior and the openness to more sustainable alternatives?

(2)　Which factors have the strongest influences

　　　(a)　on the consumption behavior and the use of clothing in general and
　　　(b)　on sustainable clothing consumption supporting behaviors?

The answers can help to understand restraints as well as to identify strategic points for a broader diffusion of more sustainable clothing consumption patterns.

The study is made for Germany, which is not only the largest clothing market in the European Union [36], but with about 12 kg per head and year also one of the three countries with the highest clothing consumption levels worldwide [37].

## 2. Materials and Methods

As a part of a wider transdisciplinary project on the question as to how sustainability could be improved in the fashion market [9], a representative population survey was conducted in Germany in 2017. It comprised computer-assisted face-to-face-interviews (CAPI) with 2000 adult, German-speaking persons, controlled for representativity with respect to age, gender, educational level, household size, and region of the place of residence. The structure of the sample is shown in Table 1.

As already stated in the introduction, it must be assumed that the clothing consumption behavior is subject to the influence of a broad spectrum of factors. In our study we suppressed possible cultural and situational factors and focused on personal and social factors as well as consumers' product-related requirements. Thus, the programmed questionnaire used by the interviewers collected the standard socio-demographic data and covered the following topics:

- personal importance of fashion and clothing;
- buying, duration of use, and reasons for the sorting of clothes;
- motives and reasons for purchase decisions;
- attractiveness of consumption alternatives;
- attitudes towards more sustainably produced and second-hand clothing;
- problem awareness related to clothing production and consumption;
- general sustainability-related attitudes.

In the representative survey, normally six-stage response scales were used. Values ranging from 1 (e.g., "I fully agree") to 6 (e.g., "I fully disagree") were assigned to the response options. The procedure for the calculation of indices and the aggregation of several items is described in the results section.

Standard bi- and multivariate statistical methods were used to analyze the expression of attitudinal and behavioral characteristics in different population groups and to examine relationships among multiple variables.

**Table 1.** Structure of the sample.

| Characteristics | Percent |
| --- | --- |
| *Gender* | |
| Female | 51.4 |
| Male | 48.6 |
| *Age* | |
| 18–29 | 17.0 |
| 30–49 | 32.0 |
| 50–65 | 28.3 |
| ≥65 | 22.7 |
| *Household net income (€)* | |
| (without 'do not know' and 'not specified') | |
| <1.000 | 11.9 |
| 1.000–1.999 | 29.5 |
| 2.000–2.999 | 27.4 |
| 3.000–3.999 | 17.2 |
| 4.000–4.999 | 5.3 |
| ≥5.000 | 4.8 |
| *Continued* | |
| *Educational attainment (German educational system)* | |
| (without 'do not know' and 'not specified') | |
| General education school leaving certificate after grade 9 (Volks-/Hauptschulabschluss, Polytechnische Oberschule) | 37.8 |
| General education school leaving certificate after grade 10 (Mittlere Reife/Realschulabschluss, Polytechnische Oberschule) | 2.6 |
| Higher education entrance qualification (Abitur, Fachabitur) | 14.7 |
| University or university of applied sciences diploma | 15.8 |
| Other | 2.1 |

## 3. Results

In the following, selected results from the representative survey are presented. In the first subsection, findings relating to attitudes and social norms that could be relevant for clothing consumption and use are presented. The second subsection gives insights into social differences with respect to quantitative and qualitative consumption-relevant behaviors. Findings regarding the relationships between attitudes and norms on the one hand and behaviors on the other hand are presented in the last subsection.

### 3.1. Attitudes and Social Norms

#### 3.1.1. Sustainability and Problem Awareness

The production and use of clothing are related to serious environmental problems. Therefore, it seems likely that fundamental attitudes towards social and ecological issues as well as the knowledge about the problems associated with clothing could have impacts on consumption and use. Before examining these relationships, some information on the occurrence of the respective attitudes in the population is given.

The representative survey showed that the level of general sustainability awareness is quite high in Germany, as measured by the degree to which the following statements were accepted by the interviewees:

- For me, an intact environment is essential for a good life (rates of agreement on the two/three highest stages of a 6-level rating scale: 73.2%/91.1%).

- It means a lot to me to live in such a way that the environment is harmed as little as possible (agreement: 69.5%/91.0%).
- It is imperative that all people in the world have similarly good living conditions (agreement: 70.5%/89.8%).
- It is important to me to behave in such a way that the lives of other people are not impaired as far possible (agreement: 72.3%/91.9%).

More than half of the interviewees are aware of the problems in connection with the production of clothing:

- Environmental pollution by the mass production of clothing (rates of classifications on the two/three highest stages of a 6-level problem rating scale: 55.9%/84.1%);
- Bad working conditions in the clothing industry (64.2%/85.8%);
- Toxic substances in clothes that could harm health of wearers (62.1%/84.3%).

The four sustainability items and the three problem items named above were aggregated to the two meta-variables 'sustainability awareness' and 'problem awareness'. Figure 1 shows the index values for these variables, calculated for each meta-variable by first averaging for each respondent the four resp. three values assigned to the single item response stages, then taking the average for all respondents of the gender, age, and household net income segment in question and dividing it by the respective mean value for all respondents. Sustainability and problem awareness are higher for women (f) than for men (m), increase comparatively strongly with age and slightly by trend with income.

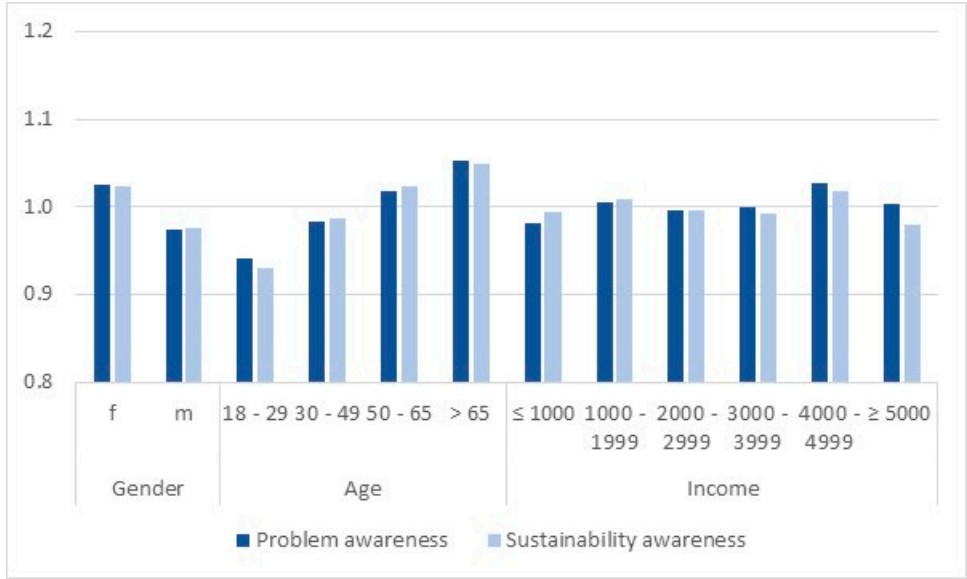

**Figure 1.** Social variability of the indices for the general sustainability awareness and the awareness of problems related to the conventional production of clothing.

### 3.1.2. Secondary Function of Clothing

Beyond its primary protective function, clothing has many secondary functions [38–44] that affect its choice, buying, and use. Clothes serve, e.g., as means to express differences to others, one's own creativity or personality. Figure 2 shows the social differentiation of the corresponding meta-variables. They are aggregates of the following items (the total population agreement rates on the two/three highest stages of a 6-level rating scale are given in parenthesis):

Distinction

- I want to show by my clothing that I belong to a certain group of people (agreement: 17.8%/38.8%).

- By my clothing, I want to set myself apart from others (agreement: 22.7%/45.1%).

Creativity

- For me it is fascinating to slip into new roles with different clothes again and again (agreement: 22.1%/44.5%).
- As for my clothing, I'm always looking for new ideas (agreement: 25.2%/51.9%).

Individuality

- With the choice of my clothes I underline my personality (agreement: 47.3%/70.0%).
- I have my own style and choose what suits me from the respective fashion (agreement: 60.8%/88.2%).
- My clothing must please myself—I don't care what others say (agreement: 70.6%/92.2%).

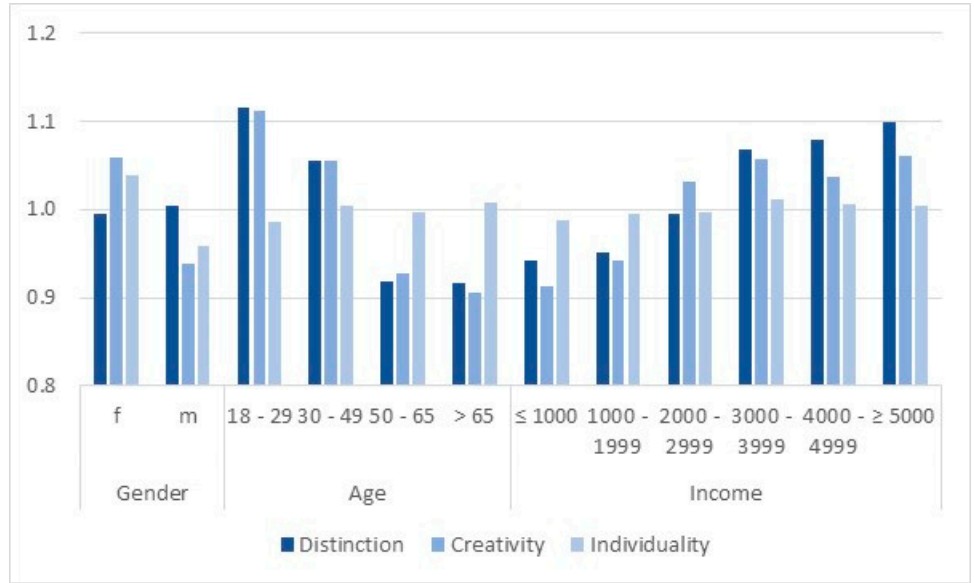

**Figure 2.** Social variability of the indices for the meta-variables 'distinction', 'creativity', and 'individuality'.

While gender does not play a role as to the importance of the distinction function of clothing, there are significant differences between women and men for the meta-variables 'creativity' and 'individuality'. The importance of the distinction function of clothing and of the creativity aspect decrease continuously with increasing age. It is widely constant for individuality, except for a slight downward deviation for the youngest population segment. For this last meta-variable, the index-values increase with income by trend. The rise is much more pronounced for the meta-variables 'distinction' and 'creativity', in the latter case, however, with a flattening of the curve for the segments with the highest incomes.

3.1.3. Consumption and Fashion

This subsection covers the results as to trendiness, consumption hedonism, and the importance of habits in the choice and the buying of clothes. Practically, it is about the following meta-variables:

Fashion orientation

- In fashion, I am often one step ahead of others (agreement: 14.7%/31.2%).
- In fashion, I know exactly what is 'in' and what is 'out' (agreement: 19.7%/44.4%).

- Usually, I wait until a new fashion comes out on top before I wear it (rates of disagreement on the two/three lowest stages of a 6-level rating scale: 17.7%/38.9%).
- I do not take part in this whole fashion rigmarole (disagreement: 9.2%/21.0%).
- When buying outerwear, it is important that it matches the current fashion trend (agreement: 27.8%/57.6%).

Consumption orientation

- Buying clothes is great fun for me (agreement: 35.8%/64.0%).
- I like to go on shopping tour with others (agreement: 24.5%/45.2%).
- I regularly clear out my wardrobe to make room for new things (agreement: 20.8%/50.6%).
- I often buy clothes that I practically do not wear anymore afterwards (agreement: 11.2%/27.8%).
- I often buy clothes impulsively, without thinking about it for a long time in advance (agreement: 41.8%/67.7%).

Habits

- I do not like changes in my clothing, I prefer to stick to my habits (agreement: 33.3%/65.9%).
- Usually, I wait until a new fashion comes out on top before I wear it (agreement: 23.5%/58.7%).

Hedonistic consumption attitudes are more pronounced for women compared to men, as Figure 3 shows. They decrease with increasing age and with income, the latter at least from lower to higher incomes. Similar patterns appear for fashion orientation, for this meta-variable, however, a continuous increase from the lowest to the highest incomes can be observed. On average the influence of habits on the buying of clothing is stronger for men than for women and it decreases with increasing income. There is no clear effect of age.

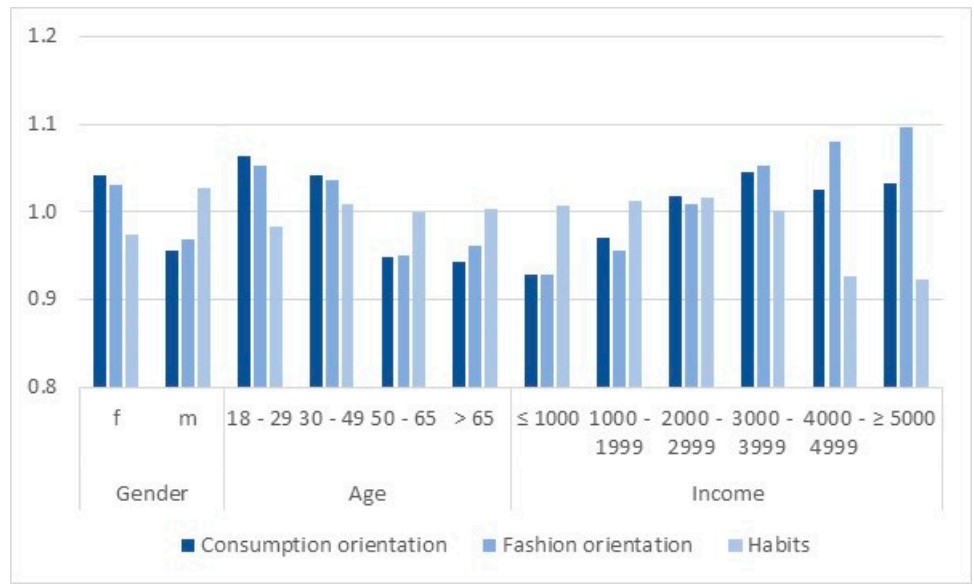

**Figure 3.** Social variability of the indices for the meta-variables 'consumption orientation', 'fashion orientation', and 'habits'.

3.1.4. Social Norms

Social norms, that is, informal rules in groups and societies that govern individual behavior, were included in the analysis for three fields of action: long wearing time of clothes, buying of more sustainably produced clothes, and buying of second-hand clothes (see Figure 4). The corresponding meta-variables comprise descriptive as well as injunctive norms:

Wearing time

- Most people who are important to me keep and wear their clothes for a long time (agreement: 38.1%/76.1%).
- Most people who are important to me would appreciate if I wear my clothes for a long time (agreement: 26.5%/64.5%).

Sustainably produced clothing

- Most people who are important to me wear sustainably produced clothing (agreement: 14.4%/41.9%).
- Most people who are important to me would appreciate if I wear sustainably produced clothing (agreement: 24.1%/58.3%).

Second-hand clothing

- Most people who are important to me wear second-hand clothing (agreement: 8.8%/22.9%).
- Most people who are important to me would appreciate if I wear second-hand clothing (agreement: 10.7%/33.5%).

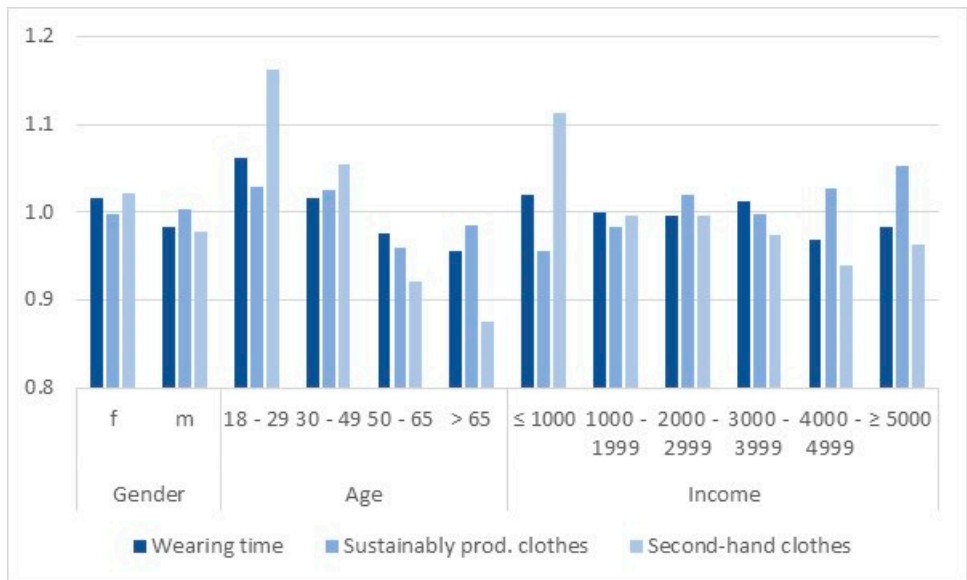

**Figure 4.** Social variability of the indices for the social norm meta-variables.

The indices for the values of the social norm meta-variables are reproduced in Figure 4 for the different population segments. The most striking and clear findings are first of all the distinct decrease of the values for the social norm meta-variable 'second-hand clothing' with age, but also the increase of the importance of the social norm related to sustainably produced clothing with income and the decrease of the values for the wearing time norm with age.

3.1.5. Price and Quality

In Figure 5 the results for two more attitudinal meta-variables are shown. These were also used in the detailed analysis of the factors that could influence the purchase and use of clothing (see below). In both cases the meta-variable comprises two variables:

Price consciousness
When buying outerwear, how important is it to you that

- it does not cost too much (rates of classifications on the two/three highest stages of a 6-level importance rating scale: 56.0%/85.6%);
- the price–performance ratio is right (97.0%/95.1%).

Quality orientation
When buying outerwear, how important is it to you that

- it consists of high-quality material (53.3%/84.1%);
- it is well made (78.5%/94.5%).

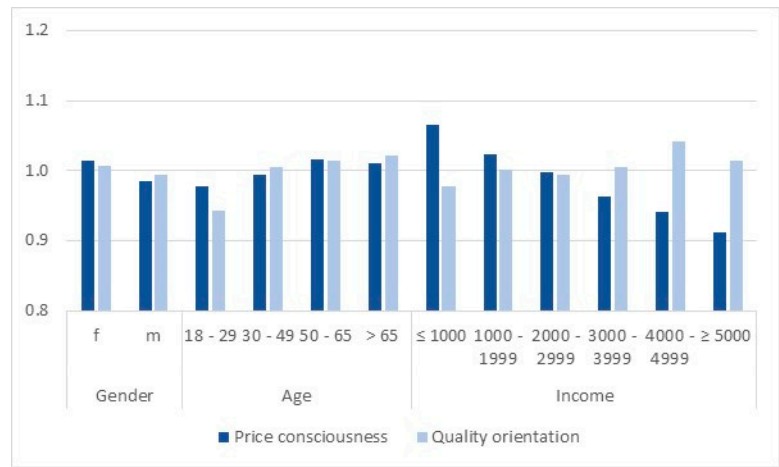

**Figure 5.** Social variability of the indices for the meta-variables 'price consciousness', and 'quality orientation'.

For both meta-variables only minor differences with respect to gender and positive correlations with age were found. As income rises, price loses importance as a buying criterion. The quality demands increase but only by trend.

*3.2. Consumption Behavior*

The survey data covered a wide range of behavioral patterns. In the following, results with respect to the consumption and the lifetime of clothing (Section 3.2.1) and behaviors that would contribute to a more sustainable clothing consumption (Section 3.2.2) are presented.

3.2.1. Consumption and Lifetime of Clothing

Figure 6 shows indices for:

- the quantitative level of outerwear consumption, calculated from the number of items bought in the last year, weighted by rough factors for the respective resource input and expenditures in manufacturing [2];
- the average wearing time for the same types of garment.

The quantitative level of clothing consumption is higher for women than for men, decreases with age and increases strongly with income. There are practically no gender differences in the wearing time of clothes. The increase by trend with age and decrease by trend with income are only slight.

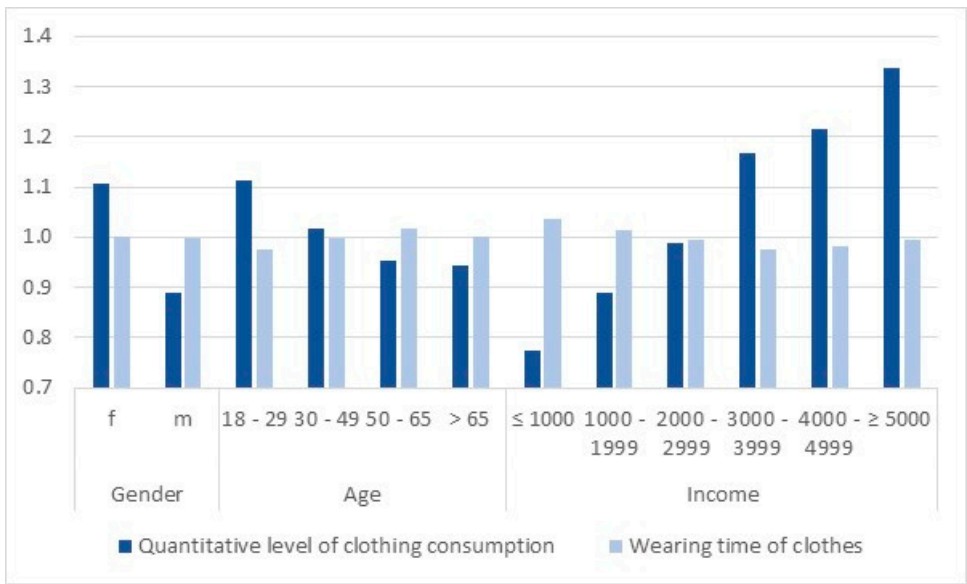

**Figure 6.** Social variability of the indices for the quantitative level of clothing consumption and the wearing time of clothes.

3.2.2. Sustainable Clothing Consumption Supporting Behaviors

Amongst others, the following three behaviors could contribute to more sustainable clothing consumption. Two of the corresponding variables sum up the answers to two questions, while the other refers to a single question:

Restriction of clothing consumption

- I try to get along with as little clothing as possible (agreement: 26.2%/55.3%).
- I only buy clothes if I really need them (agreement: 42.7%/69.4%).

Purchase of more sustainably produced clothing
How often do you use the following offers and possibilities in connection with clothing?

- buy clothing manufactured environmentally friendly (often: 9.8%, occasionally: 49.9%);
- buy clothing manufactured under fair labor conditions (often: 12.1%, occasionally: 56.2%).

Purchase of second-hand clothing
How often do you use the following offers and possibilities in connection with clothing?

- buy second-hand clothing (often: 5.9%, occasionally: 35.0%).

As to their own statements, more than half of the interviewees restrict their clothing purchase. The portion of those who often buy clothing produced under ecologically and/or socially less harmful conditions is around ten percent. The portion of second-hand buyers is even lower. The relative societal distribution of the three more sustainable consumption options can be seen in Figure 7. While the purchase of more sustainably produced or of second-hand clothing is more prevalent among women, more men stated that they restrict their clothing consumption. The index for this latter option increases by trend with age and decreases strongly with increasing income. The indices for the purchase of more sustainably produced and of second-hand closing show largely opposing trends. The buying of more sustainably produced clothing is more common in older and higher-income segments compared to younger and lower-income segments, while the purchase of second-hand clothing decreases with age and income, with the exception of the highest income segment.

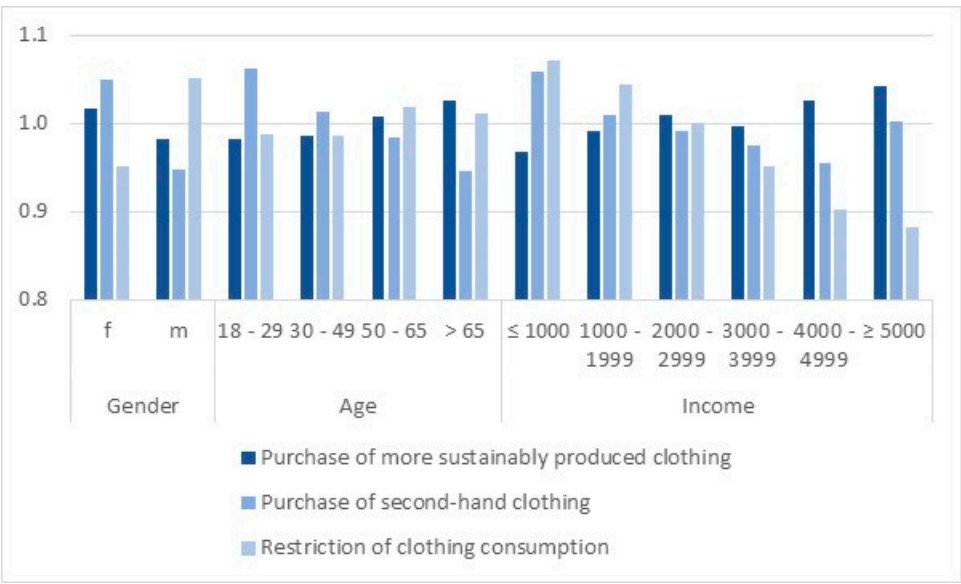

**Figure 7.** Social variability of the indices for the variables for three sustainability supporting consumption behaviors.

### 3.3. Attitudes, Norms, and Behaviors

Possible relationships between (a) the sociodemographic attributes age and income; and (b) the attitudes and norms, discussed in Section 3.1, on the one hand (left column) and the behaviors, examined in Section 3.2, on the other (first row) were tested by correlation and multiple regression analyses. The results of the zero-order correlations are given in Table 2. The levels of significance are indicated (for explanations of the symbols see footer of Table 2). The correlation coefficients are chosen for presentation since the degrees to which variances in the behavioral variables are explained by the sociodemographic-, attitude-, and norm-related variables can be directly derived by squaring the correlation coefficients given in Table 2. The results of the regression analyses confirm these findings. The coefficients were given for all respondents (column 'all') as well as separately for women (column 'f') and men (column 'm').

As already suggested by Figure 6 the quantitative level of clothing consumption correlates with income and there is a weak negative correlation with age. The strongest correlations exist for the attitudinal meta-variables 'fashion orientation', 'creativity', and 'consumption orientation'. For 'fashion orientation' there is a significant difference between women and men as to the strengths of the correlations. Another item with a greater gender difference is 'habits': For both sexes, the increasing importance of habits for the buying of clothes comes along with a decreasing level of clothing consumption. However, the negative correlation effect is stronger for women than for men.

The attitudinal factors that correlate strongly with clothing consumption are also conspicuous in connection with the wearing time of clothes. However, here the correlations are all negative and the greatest difference with respect to gender arises for 'creativity'. A growing awareness of the problems related to the production of clothing is associated with longer wearing times, but the effect appears only for women and is relatively weak.

The general sustainability awareness and clothing-related problem awareness are relatively strong predictors for the buying of more sustainably produced clothing. A highly significant correlation also exists with the perception of the corresponding social norms. An increased quality orientation also comes along with a higher importance of ecological and social buying criteria. This effect is stronger for women than for men. For the latter, there is a highly significant, albeit weak, correlation with the relevance of clothing as a means of distinction. The slight age effect seen in Figure 7 can be attributed primarily to the age-dependent buying behavior of women.

**Table 2.** Correlations of sociodemographic attributes, attitudes, and norms on the one hand and behaviors on the other hand.

| | Quantitative Level of Clothing Consumption | | | Wearing Time of Clothes | | | Purchase of More Sustainably Produced Clothing | | | Purchase of Second-Hand Clothing | | | Restriction of Clothing Consumption | | |
|---|---|---|---|---|---|---|---|---|---|---|---|---|---|---|---|
| **Gender** | all | f | m | all | f | m | all | f | m | all | f | m | all | f | m |
| **Sociodemographic attributes** | | | | | | | | | | | | | | | |
| Age | −0.08 ** | −0.10 ** | −0.06 * | 0.05 * | 0.06 * | 0.04 | 0.08 *** | 0.12 *** | 0.04 | −0.17 *** | −0.15 *** | −0.22 *** | 0.04 | 0.04 | 0.05 |
| Income | 0.21 *** | 0.22 *** | 0.23 *** | −0.08 *** | −0.07 * | −0.09 ** | 0.08 *** | 0.07 * | 0.10 ** | −0.08 *** | −0.02 | −0.10 *** | −0.17 *** | −0.15 *** | −0.23 *** |
| **Attitudes** | | | | | | | | | | | | | | | |
| Consumption orientation | 0.40 *** | 0.41 *** | 0.38 *** | −0.40 *** | −0.38 *** | −0.44 *** | 0.07 ** | 0.03 | 0.08 * | 0.15 *** | 0.08 * | 0.17 *** | −0.37 *** | −0.35 *** | −0.35 *** |
| Fashion orientation | 0.43 *** | 0.47 *** | 0.38 *** | −0.33 *** | −0.31 *** | −0.35 *** | 0.09 *** | 0.07 * | 0.09 ** | 0.06 ** | 0.00 | 0.09 ** | −0.48 *** | −0.47 *** | −0.48 *** |
| Distinction | 0.27 *** | 0.30 *** | 0.27 *** | −0.26 *** | −0.27 *** | −0.25 *** | 0.07 *** | 0.03 | 0.12 *** | 0.10 *** | 0.04 | 0.18 *** | −0.21 *** | −0.17 *** | −0.26 *** |
| Creativity | 0.41 *** | 0.41 *** | 0.38 *** | −0.31 *** | −0.28 *** | −0.36 *** | 0.09 *** | 0.04 | 0.11 *** | 0.15 *** | 0.07 * | 0.18 *** | −0.34 *** | −0.32 *** | −0.32 *** |
| Individuality | 0.24 *** | 0.23 *** | 0.21 *** | −0.09 *** | −0.07 * | −0.12 *** | 0.20 *** | 0.17 *** | 0.19 *** | 0.08 *** | 0.05 | 0.02 | −0.07 *** | −0.04 | −0.03 |
| Price consciousness | −0.09 *** | −0.07 * | −0.15 *** | 0.13 *** | 0.11 *** | 0.14 *** | −0.01 | 0.00 | −0.03 | 0.07 ** | 0.11 *** | 0.00 | 0.27 *** | 0.23 *** | 0.36 *** |
| Quality orientation | 0.17 *** | 0.15 *** | 0.20 *** | −0.10 *** | −0.08 ** | −0.11 *** | 0.30 *** | 0.33 *** | 0.26 *** | −0.06 ** | −0.06 * | −0.08 ** | 0.01 | 0.03 | 0.01 |
| Habit | −0.15 *** | −0.18 *** | −0.10 ** | 0.00 | −0.03 | 0.04 | −0.01 | −0.03 | 0.02 | −0.02 | −0.02 | 0.02 | 0.42 *** | 0.41 *** | 0.41 *** |
| Problem awareness | 0.08 *** | 0.04 | 0.09 ** | 0.04 | 0.11 *** | −0.02 | 0.31 *** | 0.30 *** | 0.31 *** | −0.02 | −0.04 | −0.07 * | −0.06 ** | −0.06 | −0,02 |
| Sustainability awareness | 0.10 *** | 0.11 *** | 0.03 | 0.06 ** | 0.09 ** | 0,03 | 0.38 *** | 0.37 *** | 0,37 *** | 0.01 | 0,04 | −0.07 * | 0.00 | 0.02 | 0,02 |
| **Social norms** | | | | | | | | | | | | | | | |
| Wearing time | ./. | ./. | ./. | 0.07 ** | 0.08** | 0.06* | ./. | ./. | ./. | ./. | ./. | ./. | ./. | ./. | ./. |
| Sustainably prod. clothes | ./. | ./. | ./. | ./. | ./. | ./. | 0.31 *** | 0.28 *** | 0.34 *** | ./. | ./. | ./. | ./. | ./. | ./. |
| Second-hand clothes | ./. | ./. | ./. | ./. | ./. | ./. | ./. | ./. | ./. | 0.46 *** | 0.48 *** | 0.44 *** | ./. | ./. | ./. |

Levels of significance: *** $p \leq 0.1\%$; ** $p \leq 1\%$; * $p \leq 5\%$

The perception of the corresponding social norms is by far the strongest predictor for the buying of second-hand clothes. There are clear differences between women and men as regards age and all attitudinal meta-variables correlating with high significance and positively with the purchase of second-hand clothing: consumption orientation, distinction, and creativity. In all cases, the effects are stronger for men than for women. This is reversed for the interrelation of price consciousness and the buying of second-hand clothes, where a weak, but highly significant, correlation can be realized for women, while there is none for men.

The behavior 'restriction of clothing consumption' is characterized by relatively strong negative correlations with the meta-variables 'consumption orientation', 'fashion orientation', and 'creativity'. A positive correlation exists with habits: People with a more habitual buying behavior concurrently tend to have a restrained buying behavior. That applies also to price consciousness, where the effect is clearly stronger for the male respondents. Effects, that are stronger for men than for women, also exist for the variables 'income' and 'distinction', both correlating negatively with a restrictive consumption behavior.

## 4. Discussion

One of the main goals of the study was to see whether there are chances for a broader diffusion of more sustainable consumption behaviors. The following discussion is about the compatibility of the three strategies supporting sustainability with actual attitudes and behaviors related to fashion and consumption.

### 4.1. Sufficiency

Against the background of the rapidly increasing demand for clothing, the depletion of resources, and the extent of the problems associated with the production of clothing, first of all it must be asked whether sufficiency is a realistic option. About one-fourth of the interviewed consumers stated with enough clearness ("I strongly agree" or "I agree") that they restrict their clothing consumption to a minimum. In many cases, however, the buying resistance is not voluntary, but because of their low income, as the data on the income effects on the quantitative level of clothing consumption (Figure 6) and on restrictive consumption behavior (Figure 7) indicate. High incomes are drivers for clothing consumption and are opposed to consumption restriction, as already stated in other studies [15–17], but attitudinal effects are even stronger, especially distinct fashion and consumption orientations. For nearly two-thirds of the respondents the buying of clothes is associated with pleasure and the conformity of clothing with the actual fashion trend is important for nearly 60%. The factors opposing a restrictive consumption behavior are particularly pronounced in the younger population segments. Against this background, it is not surprising that the probability for a reduction of the demand for clothing in Germany was rated low in an expert survey [1]. The experts even predicted that the present trend to buy more clothes at decreasing cost per garment will continue, at least for the next ten years.

### 4.2. Consistency

Up to now, the demand for more sustainably produced clothing has been low. Ecologically produced clothing is still a niche segment. The market share of clothing with the GOTS-label, that stands for strict environmental criteria, is well below one percent [45]. The same applies to clothing produced under fair labor conditions. In view of the low sales volume, the statement made by almost 10% of the interviewees that they frequently buy environmentally friendly manufactured clothing seems somewhat exaggerated. However, the general sustainability awareness and the clothing-related problem awareness are high and both, as well as the perceived social norms, are relatively strong predictors for the buying of more sustainably produced clothing. These findings are in line with studies showing a positive effect of biospheric or rather altruistic values and personal norms [30–33]. Even though other criteria, like comfort and correct fit, a good price-performance ratio, and good workmanship, often have higher weights, it is very important or at least important for almost 50% of

the customers that the clothing they buy was produced in an environmentally safe manner and under fair labor conditions [1]. In the expert survey already mentioned, the demand for more sustainable clothing is expected to increase substantially by 2030 [1]. It is also predicted that the share of sustainably produced fibers in the textile market as well as the implementation of less ecologically damaging production methods will proceed rapidly in the next decade [46–48], possibly in response to the expected evolution in demand.

*4.3. Efficiency*

As outlined in Section 1, the efficiency of material- and energy-inputs in the production of clothing can be improved indirectly by a prolonged and more intensive use of clothing. One option discussed above is the buying and use of second-hand clothing. The percentage of second-hand consumers is even lower than that of the buyers of sustainably produced clothing. This could be explained by widespread reservations towards second-hand clothing, like 'not fashionable', 'assortment too small', and 'low quality', and the bias that used clothing is something for needy persons [1,49,50]. The most important factor for the buying and wearing of second-hand clothes is the corresponding social norm. As to the chances for a broader diffusion of this consumption option, which has the potential for a substantial contribution to a more efficient use of resources, two findings are noteworthy: a relatively high level of awareness of social norms supporting the purchase of second-hand clothing (Figure 4) and a corresponding buying behavior (Figure 7) in the younger population segments. However, it should be noted that, at present, the buying of second-hand clothing is only a niche consumption variant.

Another remarkable result of the study is that the general sustainability and the problem awareness, which both reach high levels in the German population, do have positive effects on the willingness to buy more sustainably produced clothing but do not affect the buying of second-hand clothing and the restriction of clothing consumption.

## 5. Conclusions

In summary, the results of the study show that some preconditions for a broader diffusion of more sustainable alternatives in clothing consumption are given in Germany, such as a widespread sensitization for the problems associated with the conventional mass production of clothing and directly or at least subliminally perceived social norms supporting a longer use of clothing and, to a lesser extent, the buying of more sustainably produced and second-hand clothing. It is not unimportant that this especially applies, in different manners, to younger and well-established population segments, since both fulfill certain societal orientation functions regarding lifestyle, and both represent important customer segments. However, due to strong fashion and consumption orientations as well as high levels of consumption, both groups also represent consumers who are problematic from a sustainability point of view.

So far, the function of the well-established and parts of the younger consumer segments as role models for others has been counterproductive. This and their high levels of consumption make them important target groups for sustainable consumption strategies in the field of clothing. Even knowing that a substantial reduction of the quantitative level of consumption is indispensable as regards these target groups, one must, as a start, content oneself with an attenuation of the negative impacts of the production and use of clothing on humans, the environment, and resources. That means to push the demand for more efficient and ecologically as well as socially acceptable alternatives. This cannot be achieved by product marketing alone but requires target group adequate social-marketing strategies. In the well-established population segments, the relatively high sustainability awareness, the efficacy of the corresponding social norms, and target group-compatible shining examples should be used to strengthen the demand for clothing produced under ecologically and socially acceptable conditions. The fashion- and creativity-oriented younger population segments should be addressed to accelerate the diffusion of consumption variants supporting the multiple and thus more efficient uses of clothing,

like the buying of second-hand and the swapping of clothes. This requires target group adequate measures to overcome negative preconceptions as to second-hand clothing.

Beyond the strengthening of the demands for more sustainably produced and second-hand clothing and of collaborative forms of clothing consumption, a strict and reliable regulatory framework is needed that supports a shift of demand and a transformation of the whole textile industry towards ecologically and socially acceptable production conditions.

However, as already emphasized, the ultimate goal must be to substantially reduce the extent of clothing consumption. However, given the complex secondary functions of clothing, this requires not less than a fundamental cultural change.

**Author Contributions:** All authors contributed equally to the research and to the preparation of the manuscript.

**Funding:** This research was funded by the German Federal Ministry of Education and Research.

**Acknowledgments:** The authors thank their colleagues in the InNaBe-project for inspiring discussions and Nora Schmidt for assistance in data processing. Valuable suggestions by the editor and the reviewers are gratefully acknowledged.

**Conflicts of Interest:** The authors declare no conflict of interest.

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
