# Peer review of "Social Acceptability of More Sustainable Alternatives in Clothing Consumption"

_sustainability, doi:10.3390/su11226194_

Round 1

Reviewer 1 Report

Thank you for the opportunity to review your paper entitled “preconditions and chances for the diffusion of more sustainable alternatives in clothing consumption”. Whilst the topic is interesting it is unclear what this article seeks to achieve. The main concerns are a lack of awareness of what has already been investigated in the literature and the method of analysis.

Introduction:

interesting ideas brought forward, yet unclear what sustainability is, this needs to be carefully defined unclear why the authors highlight what the three strategies are that contribute to improve sustainability in production and consumption and the textile chain efficiency if they are focusing on the attitude behaviour gap Why is this research important? What is the reader learning from reading this paper? What is the aim and objectives?

Literature Review

this part is missing completely and needs to be included to demonstrate how the current study is extending knowledge

Materials & Methods

What is a representative population survey? Why Germany? Where 2000 face-to-face interviews conducted? This seems a very quantitative size and surly saturation point would have been hit prior to this number? It is confusing, was this a questionnaire or face-to-face interviews? Table 1 – it would be beneficial to translate the German terms or provide equivalents, e.g. Abitur or Fachabitur are A-level equivalents

Results:

what was measured in the first place and why? The analysis seems rather simplistic and it is not clear why the data was analysed in this manner

Author Response

We thank the reviewer for valuable questions and hints. These have been considered in the revision as follows:

Thank you for the opportunity to review your paper entitled “preconditions and chances for the diffusion of more sustainable alternatives in clothing consumption”. Whilst the topic is interesting it is unclear what this article seeks to achieve. The main concerns are a lack of awareness of what has already been investigated in the literature and the method of analysis.

The research questions are given explicitly now.
The literature overview in the introductory paragraph has been expanded considerably.

Introduction:

interesting ideas brought forward, yet unclear what sustainability is, this needs to be carefully defined unclear why the authors highlight what the three strategies are that contribute to improve sustainability in production and consumption and the textile chain efficiency if they are focusing on the attitude behaviour gap Why is this research important? What is the reader learning from reading this paper? What is the aim and objectives?

Sustainability in the context of clothing is defined. The description of the theoretical background has been expanded. The passage on clothing production has been deleted. The three strategies are the read threat of the article (see discussion). We do not focus on the attitude behaviour gap but emphasize that problem and sustainability awareness correlate positively with the buying of more sustainably produced clothing but not with other more sustainable consumption options.
The research questions are given explicitly now.

Literature Review

this part is missing completely and needs to be included to demonstrate how the current study is extending knowledge

see above

The literature with relevance for the study is recognized.

Materials & Methods

What is a representative population survey? Why Germany? Where 2000 face-to-face interviews conducted? This seems a very quantitative size and surly saturation point would have been hit prior to this number? It is confusing, was this a questionnaire or face-to-face interviews? Table 1 – it would be beneficial to translate the German terms or provide equivalents, e.g. Abitur or Fachabitur are A-level equivalents

Additional methodical information is given.
The high number of interviewees was needed to guarantee statistical significance even for small subgroups and/ or small effects.
The method was Computer Assisted Personal Interviews, that is the interviewer used a questionnaire on his tablet to put the question (in randomized form if necessary) and to record the answers.
The choice of the research region is substantiated.
The international or Anglo-Saxonian equivalents of the German graduations are given.

Results:

what was measured in the first place and why? The analysis seems rather simplistic and it is not clear why the data was analysed in this manner

The questions and the used scales are given. The proceeding of the data from the interviewees answers to the indexes is described.

Reviewer 2 Report

The paper deals with many interesting and interesting topics.
Following are the observations:
- on page 3, translating the German parts of the qualification into English,
paragraph 3 (as a whole) is difficult to read, is very schematic, I think it should be made more fluid in order to better appreciate its content, I propose a substantial reorganization,
- I advise you to modify figures 1 to 7, so they are not immediately understandable,
increasing and diversifying the references, 5/22 are the authors.

Author Response

We thank the reviewer for valuable suggestions. These have been considered in the revision as follows:

The paper deals with many interesting and interesting topics.
Following are the observations:

- on page 3, translating the German parts of the qualification into English,

The international or Anglo-Saxonian equivalents of the German graduations are given.

paragraph 3 (as a whole) is difficult to read, is very schematic, I think it should be made more fluid in order to better appreciate its content, I propose a substantial reorganization,

In the results chapter we limited ourselves strictly to the presentation of the findings. The classification and appreciation of the results follow in the discussion paragraphs.

- I advise you to modify figures 1 to 7, so they are not immediately understandable,

The type of presentation was chosen to allow a direct comparison of the results for related subjects as regards contents (attitudes, behaviours). A splitting of the figures would hinder the comparison.

increasing and diversifying the references, 5/22 are the authors.

The literature overview in the introductory paragraph has been expanded considerably.

The literature with relevance for the study is recognized.

Reviewer 3 Report

The paper is about an interesting problem to be solved, that is, sustainable alternatives in clothing consumption. It presents a set of original findings that need further development and improvement, for future publication. Although I have a favourable vision, there are several aspects that need to improve in a new version of the manuscript, namely:

The introduction needs to be substantially improved, especially by rewriting and summarizing the, as well as by positioning the current study whiting the ongoing debate on the sustainability of clothes’ consumption, using references from your target journal. In the same introductory item, it is suggested to clearly outline the goals and contributions of the study into the emerging literature of sustainable business models. Adding to this, a paragraph presenting the structure of the paper is required, at the final of the introductory item. The manuscript is lacking theoretical support, it is suggested to create a new item titled: 2. Theoretical framework; where the previously referred triple approach on efficiency, consistency and sufficiency could be the guiding cornerstone. In the Discussion item, it is suggested to maintain the previous sequence: efficiency, consistency and sufficiency. In the Conclusions' item, in light of the results obtained, several implications could be made available to policymakers and responsible managers of the clothing industry. Please remove the item 7. Declarations.

Considering the previous comments, and although I recognize potential to the ongoing research, I recommend a major revision of the manuscript.

Author Response

We thank the reviewer for valuable suggestions. These have been considered in the revision as follows:

The paper is about an interesting problem to be solved, that is, sustainable alternatives in clothing consumption. It presents a set of original findings that need further development and improvement, for future publication. Although I have a favourable vision, there are several aspects that need to improve in a new version of the manuscript, namely:

The introduction needs to be substantially improved, especially by rewriting and summarizing the, as well as by positioning the current study whiting the ongoing debate on the sustainability of clothes’ consumption, using references from your target journal.

The introduction has been revised comprehensively.
The literature review in the introductory paragraph has been expanded considerably.

In the same introductory item, it is suggested to clearly outline the goals and contributions of the study into the emerging literature of sustainable business models.

Another reviewer called for a concentration on the consumer perspective. The work on business models was done by another partner in our consortium. The results are published elsewhere.

Adding to this, a paragraph presenting the structure of the paper is required, at the final of the introductory item.

The basic structure is predetermined by the editor resp. the journal. Information on the structure of the presentation of the findings is given at the beginning of the results chapter.

The manuscript is lacking theoretical support, it is suggested to create a new item titled: 2. Theoretical framework; where the previously referred triple approach on efficiency, consistency and sufficiency could be the guiding cornerstone.

The description of the theoretical background has been expanded.

In the Discussion item, it is suggested to maintain the previous sequence: efficiency, consistency and sufficiency.

The succession in the discussion was chosen to accentuate the importance of sufficiency for more sustainability in clothing consumption. In the introduction we followed the historical development of the sustainability concept.

In the Conclusions' item, in light of the results obtained, several implications could be made available to policymakers and responsible managers of the clothing industry.

The conclusions have been extended. But, since the production related statements in the introduction have been deleted due to the focussing on the consumer perspective, the conclusions as to the need for actions on the side of the industry and for an effective regulatory framework are only given in a general way.

Please remove the item 7. Declarations.

This paragraph is given by the journal.

Considering the previous comments, and although I recognize potential to the ongoing research, I recommend a major revision of the manuscript.

Round 2

Reviewer 1 Report

Thank you for making the improvements as suggested, the article has a better flow to it now. All sections have been improved and satisfy the comments. Only aspect that the authors may want to consider is adding slightly more detail on the method of analysis

Author Response

Answer to reviewer

We thank you for reviewing our revised manuscript. We added a short passage in the methods section.

Reviewer 2 Report

I appreciated the effort and commitment of the authors who improved the quality of the paper.
I am satisfied with the interventions made on the text.

Author Response

Answer to reviewer

We thank you for reviewing our revised manuscript.

Reviewer 3 Report

The authors addressed in a satisfactory way the previous suggestions. Thus, I recommend the acceptance of the manuscript, although additional editorial care is required before publication, especially in the Introductory item.

Author Response

Answer to reviewer

We thank you for reviewing our revised manuscript. We submitted a text with our changes accepted. But it seems that you obtained our revised manuscript in a text-comparing version showing the changes made. We suppose that your comment as to the editorial care required in the introductory part refers to the bad readability of this part in that version. Or do you have concrete suggestions for the improvement of the introductory part.

We added a short passage in the methods section.